# Observation of flat band, Dirac nodal lines and topological surface states in Kagome superconductor CsTi$_3$Bi$_5$

Jiangang Yang[1,2,8], Xinwei Yi [2,8], Zhen Zhao[1,2,8], Yuyang Xie[1,2,8], Taimin Miao[1,2], Hailan Luo [1,2], Hao Chen[1,2], Bo Liang[1,2], Wenpei Zhu[1,2], Yuhan Ye[1,2], Jing-Yang You[3], Bo Gu[2,4,5], Shenjin Zhang[6], Fengfeng Zhang[6], Feng Yang[6], Zhimin Wang[6], Qinjun Peng[6], Hanqing Mao[1,2,7], Guodong Liu[1,2,7], Zuyan Xu[6], Hui Chen [1,2,4], Haitao Yang [1,2,4], Gang Su [2,4,5] ✉, Hongjun Gao [1,2,4] ✉, Lin Zhao [1,2,7] ✉ & X. J. Zhou [1,2,7] ✉

Kagome lattices of various transition metals are versatile platforms for achieving anomalous Hall effects, unconventional charge-density wave orders and quantum spin liquid phenomena due to the strong correlations, spin-orbit coupling and/or magnetic interactions involved in such a lattice. Here, we use laser-based angle-resolved photoemission spectroscopy in combination with density functional theory calculations to investigate the electronic structure of the newly discovered kagome superconductor CsTi$_3$Bi$_5$, which is isostructural to the AV$_3$Sb$_5$ (A = K, Rb or Cs) kagome superconductor family and possesses a two-dimensional kagome network of titanium. We directly observe a striking flat band derived from the local destructive interference of Bloch wave functions within the kagome lattice. In agreement with calculations, we identify type-II and type-III Dirac nodal lines and their momentum distribution in CsTi$_3$Bi$_5$ from the measured electronic structures. In addition, around the Brillouin zone centre, $\mathbb{Z}_2$ nontrivial topological surface states are also observed due to band inversion mediated by strong spin-orbit coupling.

Quantum materials with layered kagome structures have drawn enormous attention because such a two-dimensional (2D) network of corner-sharing triangle lattice can give rise to many exotic quantum phenomena, such as spin liquid phases[1–4], topological insulator, semimetal and superconductor[5–9], fractional quantum Hall effect[10], quantum anomalous Hall effect[11–14] and unconventional density wave orders[15–20]. All these exotic quantum phenomena are thought to originate from the unique electronic structure of the kagome lattice including flat bands, Dirac cones and saddle points when the spin-orbit coupling, magnetic ordering or strong correlation are taken into consideration. Nevertheless, the definitive identification of such unique electronic structures in the kagome materials is still scarce and the underlying mechanism to induce those exotic quantum phenomena from such electronic structures remains elusive. For example, the kagome superconductors AV$_3$Sb$_5$ (A=K, Rb or Cs)[21], which have been the focus of recent extensive

[1]Beijing National Laboratory for Condensed Matter Physics, Institute of Physics, Chinese Academy of Sciences, Beijing 100190, China. [2]School of Physical Sciences, University of Chinese Academy of Sciences, Beijing 100049, China. [3]Department of Physics, Faculty of Science, National University of Singapore, Singapore 117551, Singapore. [4]CAS Center for Excellence in Topological Quantum Computation, University of Chinese Academy of Sciences, Beijing 100190, China. [5]Kavli Institute of Theoretical Sciences, University of Chinese Academy of Sciences, Beijing 100190, China. [6]Technical Institute of Physics and Chemistry, Chinese Academy of Sciences, Beijing 100190, China. [7]Songshan Lake Materials Laboratory, Dongguan, Guangdong 523808, China. [8]These authors contributed equally: Jiangang Yang, Xinwei Yi, Zhen Zhao, Yuyang Xie. ✉e-mail: gsu@ucas.ac.cn; hjgao@iphy.ac.cn; lzhao@iphy.ac.cn; XJZhou@iphy.ac.cn

research, exihibit anomalous Hall effect[22,23], unconventional charge density wave (CDW)[19,24–39], pairing density wave[18] and possible unconventional superconductivity and nematic phase[17,22,23,30,40–42]. However, the nature and origin of these novel physical properties are still in hot debates. Even for the clear identification of the flat band, it still needs further investigations. It is significant to establish a relationship between the unique electronic structures of the kagome lattice and its novel quantum phenomena.

$CsTi_3Bi_5$ is a newly discovered kagome superconductor which is isostructural to the $AV_3Sb_5$ superconductors (Fig. 1a)[43]. The titanium atoms form a kagome network with the bismuth atoms lying in the hexagons and above and below the triangles (Fig. 1a, b). Magnetic susceptibility and electrical resistivity measurements of $CsTi_3Bi_5$ indicate that there is no phase transition observed down to the superconducting transition at 4.8 K[44]. This is different from $CsV_3Sb_5$ that exhibits a CDW transition around 94 K[21]. The similar crystal structure but the absence of the CDW order in $CsTi_3Bi_5$ provide a good opportunity to study the intrinsic electronic structure of the kagome lattice with reference to $AV_3Sb_5$ and understand the origin of various quantum phenomena in kagome materials.

In this paper, we investigate the electronic structure of the newly discovered kagome superconductor $CsTi_3Bi_5$. By using high resolution laser-based angle-resolved photoemission spectroscopy (ARPES), in combination with the band structure calculations, we have directly observed the characteristic electronic features of the kagome lattice. We directly observed the flat band derived from the destructive interferences of the Bloch wave functions within the kagome lattices. We also identified the Dirac nodal loops and nodal lines in three-dimensional momentum space from the band structure calculations and the measured electronic structures are consistent with the calculated results. The $\mathbb{Z}_2$ nontrivial topological surface states are also observed. Such coexistence of multiple nontrivial band structures in one kagome superconductor provides a platform to study the rich physics in the kagome lattice.

## Results

Figure 1 d shows the Fermi surface mapping of $CsTi_3Bi_5$ measured at 20 K. The entire first BZ is covered by our laser ARPES measurements. Five Fermi surface sheets are clearly observed, as quantitatively shown in Fig. 1e. The Fermi surface consists of three electron-like Fermi surface sheets around $\bar{\Gamma}$ ($\alpha$, $\beta$ and $\gamma_1$ in Fig. 1e), an electron-like triangular Fermi pocket around $\bar{K}$ ($\gamma_2$ in Fig. 1e) and a small hole-like Fermi pocket around $\bar{M}$ ($\delta$ in Fig. 1e).

In order to understand the measured electronic structure, we carried out detailed band structure calculations. Figure 1f and g show the calculated band structures of $CsTi_3Bi_5$ without considering the spin-orbit couping (SOC) (Fig. 1f) and considering SOC (Fig. 1g). These calculations project the band structures onto different Ti 3d orbitals along high symmetry directions in the BZ. The low energy bands are mainly from the 3d orbitals of titanium. The characteristic electronic features of a kagome lattice, including the flat band, two saddle points at $\bar{M}$ and a Dirac point at $\bar{K}$, can be clearly observed as marked in Fig. 1f and g. These features are mainly from the Ti $3d_{x^2-y^2/xy}$ orbitals (red lines in Fig. 1f, g) except that the van Hove singular (VHS) point, namely the saddle point VHS1, is from Ti $3d_{z^2}$ (orange line in Fig. 1f, g). The consideration of SOC shows little effect on the flat band and the saddle points but opens a gap at the Dirac points (Fig. 1g).

The calculated band structures of $CsTi_3Bi_5$ (Fig. 1f, g) are very similar to that of $CsV_3Sb_5$ where the kagome lattice related electronic features are mainly from the V 3d orbitals[17,34]. The main difference is the band position with respect to the Fermi level. In $CsTi_3Bi_5$ the kagome lattice related bands are shifted upwards by ~1 eV when compared with those in $CsV_3Sb_5$. This is because $CsTi_3Bi_5$ has one electron less per Ti per unit cell than that of $CsV_3Sb_5$ when Ti is replaced by V. As a result, although the Ti 3d orbitals still dominate the density of states (DOS) around the Fermi level $E_F$, the two van Hove singularities in $CsTi_3Bi_5$ are above the Fermi level whereas they are close or below the Fermi level in $CsV_3Sb_5$[35,36]. This provides a possible explanation of the absence of the CDW order in $CsTi_3Bi_5$. The

**Fig. 1 | Fermi surface and calculated band structures of $CsTi_3Bi_5$. a** Schematic pristine crystal structure of $CsTi_3Bi_5$. **b** Top view of the crystal structure with a two-dimensional kagome lattice of titanium. **c** Three-dimensional Brillouin zone with high-symmetry points and high-symmetry momentum lines marked. **d** Fermi surface mapping of $CsTi_3Bi_5$ measured at a temperture of 20 K. It is obtained by integrating the spectral intensity within 10 meV with respect to the Fermi level and symmetrized assuming six-fold symmetry. Five Fermi surface sheets are clearly observed and quantitatively shown in **e**. Three Fermi surface sheets are around the Brillouin zone center $\Gamma$ marked as $\alpha$ (orange line), $\beta$ (green line) and $\gamma_1$ (light blue line). One Fermi surface is around the K point marked as $\gamma_2$ (blue line) and one is

around the M point marked as $\delta$ (dark blue line). **f** Calculated band structure along high-symmetry directions without considering SOC. Different colors represent different orbital components of Ti$_{3d}$. **g** Same as (**f**) but considering SOC. The flat band (FB), two saddle points (VHS1 and VHS2) and a Dirac point (DP) are marked by arrows. To make a better comparison with measured results, the Fermi level referred to as $E_F$(Exp.) is shifted downwards by 90 meV, as shown by the dashed lines in (**f**, **g**). **h** Calculated three-dimensional Fermi surface based on the first principle DFT calculations. The Fermi surface sheets are quite two dimensional. The calculated Fermi surface at $E_F$(Exp.) and $k_z$ = 0 is shown in (**i**). The measured Fermi surface (**d**) shows an excellent agreement with the calculated one (**i**).

upward band shift also moves the flat band close to Fermi level in CsTi₃Bi₅.

Figure 1h shows the calculated Fermi surface in three-dimensional Brillouin zone. The Fermi surface consists of five sheets which are quite two dimensional. This is expected due to the strong in-plane bonding and weak interlayer coupling in CsTi₃Bi₅ which is similar to that in CsV₃Sb₅. The calculated Fermi surface at $k_z = 0$ is shown in Fig. 1i. To make a better comparison between the measured Fermi surface and band structures with the band structure calculations, we find that the Fermi level of the calculated band structures needs to be shifted downwards by ~ 90 meV, as shown in Fig. 1f, g. The calculated Fermi surface (Fig. 1h and i) shows an excellent agreement with the measured results in Fig. 1d and e.

## Nontrivial flat band

For the genuine kagome lattice, it shows a perfect flat band across the entire Brillouin zone, as schematically shown in Fig. 2g. In real kagome materials like CsTi₃Bi₅, as shown in Fig. 1f, g, the flat band (FB) is nearly dispersionless along $\bar{K}$-$\bar{M}$-$\bar{K}$ but becomes dispersive along $\bar{\Gamma}$-$\bar{M}$ and $\bar{\Gamma}$-$\bar{K}$ directions. This is because, in real kagome materials, the flat band dispersion can be modified by additional factors besides the spin-orbit coupling, such as the in-plane next-nearest-neighbor hopping, the interlayer coupling or the multiple orbital degrees of freedom. So far there have been some ARPES studies reporting the observation of the kagome lattice-derived flat band in GdV₆Sn₆[45], YMn₆Sn₆[46], CoSn[47–49], Fe₃Sn₂[50] and FeSn[51]. However, there is little clear evidence reported about the kagome-derived flat band in the 135 family represented by AV₃Sb₅(A = K, Rb or Cs)[35,52].

Figure 2a, b show the band structures of CsTi₃Bi₅ measured at 20 K along the $\bar{\Gamma}$-$\bar{M}$-$\bar{K}$-$\bar{\Gamma}$ high symmetry directions under the LV (Fig. 2a) and LH (Fig. 2b) light polarizations. In order to resolve all the band structures more clearly, the corresponding second derivative image is shown in Fig. 2c. There seems to be a dispersionless band across the entire Brillouin zone at the binding energy of ~ 0.25 eV. A careful analysis indicates that this band consists of two different parts. The first part is marked by the red dashed line in Fig. 2c while the rest of the band represent the second part. As compared with the band structure calculations in Fig. 2d, e, the first part shows a good agreement with the flat band from the band structure calculations. This demonstrates that it is kagome lattice derived flat band. This band can be attributed to the local destructive interferences of the Bloch wave functions within the kagome lattices (Fig. 2f), as described in Supplementary Materials. The second part of the dispersionless band at ~0.25 eV is not expected from the band structure calculations (Fig. 2d, e). A careful inspection indicates that there is an additional spectral weight buildup in the binding energy range of 0.25 ~ 0.50 eV. This spectral weight buildup is bounded by two cutoffs: the upper bound at −0.25 eV and the lower bound at −0.50 eV. The second derivative (Fig. 2c) makes the two boundaries show up like two bands but they are not the real part of the band structure. Therefore, the second part flat band at ~0.25 eV actually represents a spectral weight cutoff at this energy (see Supplementary Fig. 3 for details).

The observation of high intensity buildup between [−0.25, −0.50] eV in CsTi₃Bi₅ is an unexpected finding. It is not expected from the band structure calculations. To the best of our knowledge, such a spectral buildup over the entire Brillouin zone has not been observed by ARPES in other materials. One possibility to check is whether this may be attributed to the $k_z$ effect. Due to finite $k_z$ resolution, the measured band structure may correspond to the summation of bands at different $k_z$s. The measured data are found to be not consistent with the calculated results ($k_z = 0 \sim 1$), indicating that the $k_z$ effect is unlikely (Supplementary Fig. 4). The two cutoff energies of the spectral weight buildup happen to coincide with the top and bottom energy positions of the kagome derived flat band. This suggests that the extra spectral

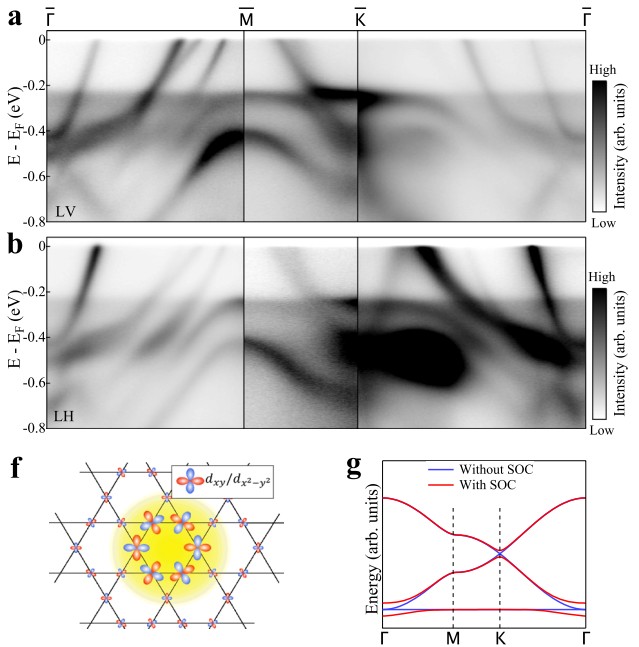

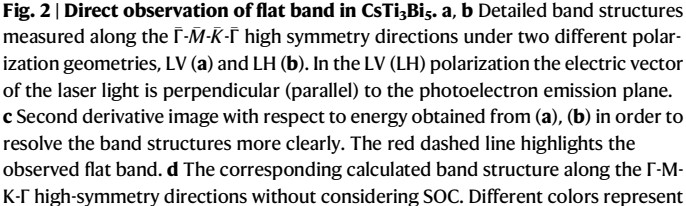

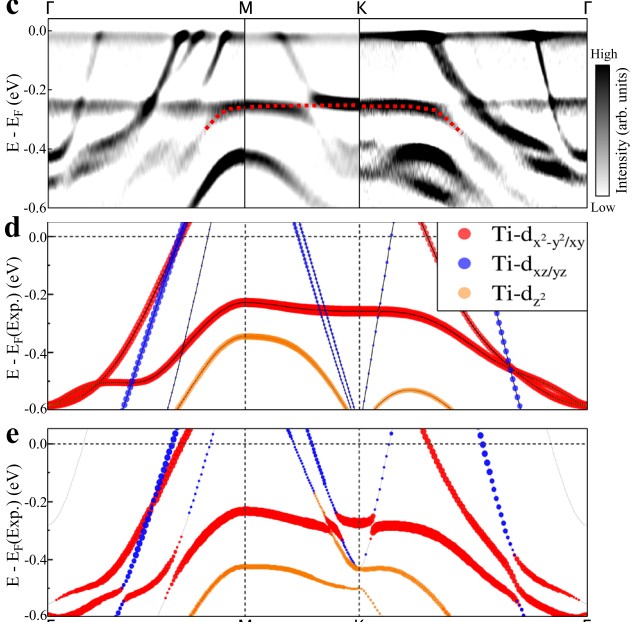

**Fig. 2 | Direct observation of flat band in CsTi₃Bi₅. a, b** Detailed band structures measured along the $\bar{\Gamma}$-$\bar{M}$-$\bar{K}$-$\bar{\Gamma}$ high symmetry directions under two different polarization geometries, LV (**a**) and LH (**b**). In the LV (LH) polarization the electric vector of the laser light is perpendicular (parallel) to the photoelectron emission plane. **c** Second derivative image with respect to energy obtained from (**a**), (**b**) in order to resolve the band structures more clearly. The red dashed line highlights the observed flat band. **d** The corresponding calculated band structure along the Γ-M-K-Γ high-symmetry directions without considering SOC. Different colors represent different orbital components of Ti₃d. To make a better comparison with measured results, the Fermi level referred to as $E_F$(Exp.) is shifted downwards by 90 meV. **e** Same as (**d**) but considering SOC. **f** Orbital textures of the effective Wannier states giving rise to the flat bands with $d_{xy}/d_{x^2-y^2}$ orbitals. **g** Tight-binding band structures of kagome lattice with (red lines) and without (blue lines) SOC. Inclusion of the spin-orbit coupling gaps both the Dirac crossing at K and the quadratic touching between the flat band and the Dirac band around Γ.

weight buildup is closely related to the existence of the flat band. It is possible that the spectral buildup may come from electron scattering of the flat band states. Since the flat band corresponds to high density of states confined by the band top ($\sim$−0.25 eV) and the band bottom ($\sim$−0.50 eV), the scattered electrons may lie in the same energy range. At present, we can not fully pin down the origin of the high intensity buildup whether it is an extrinsic effect or it represents an intrinsic effect due to other origins. Further efforts are needed to fully understand this interesting phenomenon.

### Dirac nodal lines

In some quantum materials, the bands can cross at a discrete point in the momentum space, forming Dirac point with spin degeneracy or Weyl point with spin polarization. The Dirac points can also form nodal lines and nodal loops in three-dimensional momentum space[53,54]. The Dirac points can be categorized into three types according to the slopes of the involved bands[55,56]. The materials, which have the electronic structure with the type-II (two dispersion branches exhibit the same sign of slope) or type-III (one of two branches is dispersionless) Dirac points, may host exotic properties, e.g., the chiral anomaly[55] and Klein tunneling[57]. However, there have been few established cases of the type-II and type-III Dirac point realization in real materials, not to mention their simultaneous observation in one material.

Figure 3a, b show our identification of two sets of Dirac nodal loops and one set of Dirac lines in CsTi$_3$Bi$_5$. Fig. 3a shows the calculated

band structure along the high-symmetry directions without considering spin-orbit coupling. We can find two groups of linear dispersion crossings in a covered energy region around $E_F$, marked as NL1 and NL2 in Fig. 3a. Our DFT calculations (Supplementary Fig. 5) reveal that these Dirac nodes are not isolated, but form multiple nodal loops in $k_z = 0$ and $k_z = \pi/c$ planes as seen in Fig 3b. The NL1 type-II nodal loops form in-plane hexagons centered on Γ and A while the NL2 nodal loops form in-plane triangles centered on all the K and H points. These nodal loops are protected by the $M_z$ mirror symmetry. Detailed band analysis shows that the type-II NL1 of $k_z = 0$ and $k_z = \pi/c$ planes are not connected along the $k_z$ direction due to the absence of the $M_z$ mirror symmetry between $0 < k_z < \pi/c$. However, for the NL2 nodal loops in the $k_z = 0$ and $k_z = \pi/c$ planes, we find another set of nodal lines in the Γ-K-H-A plane that links them. These nodal lines are type-III and protected by the $M_x$ mirror symmetry. Due to the six-fold rotational symmetry, there are six nodal lines and NL2 loops that are symmetrically distributed near K and L points. Slices at different $k_z$ positions have similar band structures, which makes these nodal loops in different slices still possible to be captured experimentally in spite of the opening of small gaps. Moreover, after considering SOC, these nodal loops will further open gaps but the gap size remains small (< 50 meV).

Figure 3c−e show the measured band structures along Γ̄-M̄, M̄-K̄ and K̄-Γ̄ high-symmetry directions, respectively. For comparison, the corresponding calculated band structures with SOC are presented in Fig. 3f−h. The calculated bands agree very well with the experimental

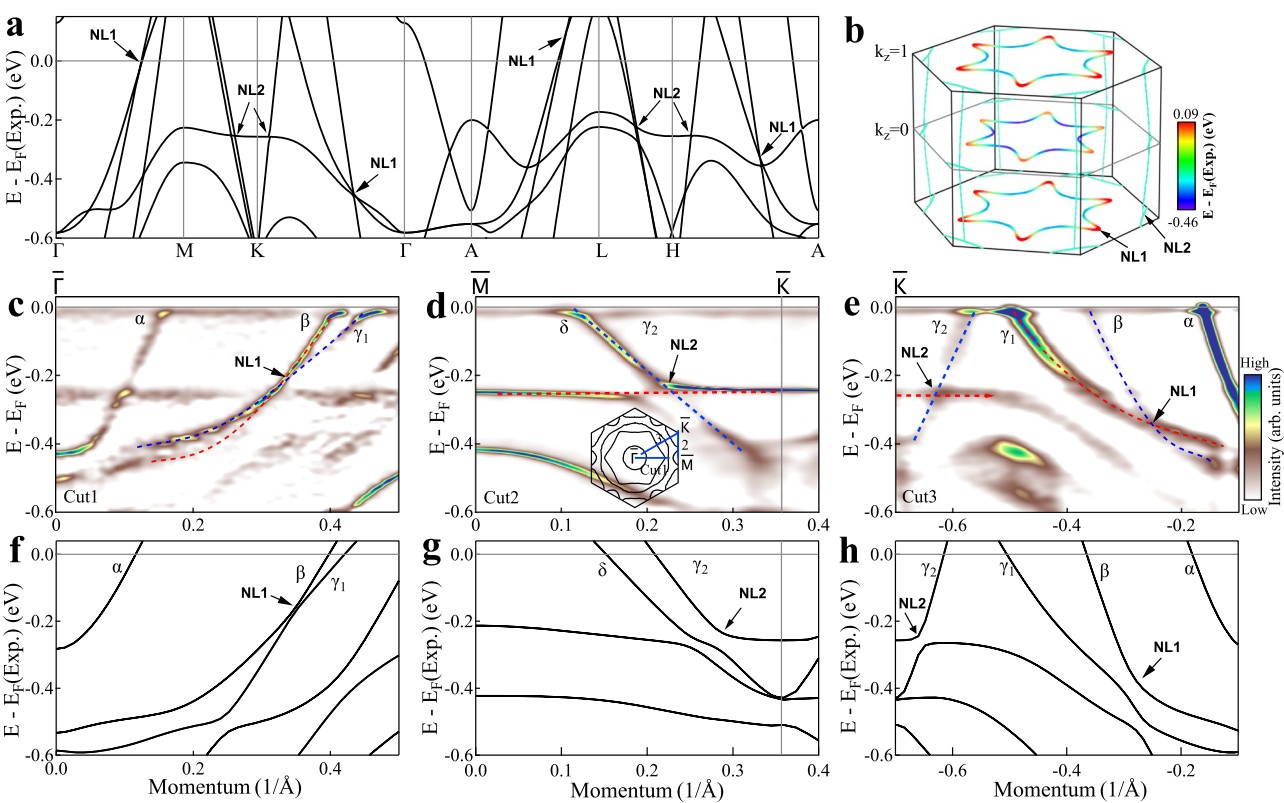

**Fig. 3 | Calculated and observed Dirac nodal points in CsTi$_3$Bi$_5$. a** Calculated band structures along high-symmetry directions without considering SOC. Two sets of Dirac nodal points (NL1 and NL2) are identified as marked by arrows. **b** Formation of the Dirac nodal lines in three-dimensional momentum space. The NL1 Dirac points form hexagonal Dirac nodal loops around Γ in the $k_z = 0$ plane and A in the $k_z = 1$ plane. The NL2 Dirac points form triangular Dirac nodal loops around K in the $k_z = 0$ plane and H in the $k_z = 1$ plane as well as six nodal lines along the $k_z$ direction. The detailed distribution of the NL1 and NL2 Dirac nodal lines in three-dimensional momentum space is shown in Supplementary Fig. 5 in Supplementary Information. **c-e** Band structures measured along Γ̄-M̄ (Cut1), M̄-K̄ (Cut2) and K̄-Γ̄

(Cut3), respectively. All three panels share the same colorbar. The location of the momentum cuts is shown in the inset of **d** by the blue lines. These images are obtained by taking second derivative curvature of the original data. The blue and red dashed lines are guide lines of the two crossing bands. We note that the non-dispersive branch at $\sim$−0.25 eV in **c** is from the cutoff of the spectral weight buildup. The features at $\sim$−0.25 eV in (**d**, **e**) represent real bands as can be seen and analyzed from the original data (Fig. 2a, b). **f–h** The corresponding calculated band structures with SOC. The presence of the Dirac points is marked by arrows. To make a better comparison with measured results, the Fermi level referred to as $E_F$(Exp.) is shifted downwards by 90 meV.

results. The NL1 point is formed by the crossing of the $\beta$ and $\gamma_1$ bands, as shown by the blue and red dashed lines in Fig. 3c, e. These two bands ($\beta$ and $\gamma_1$) share the same sign of slope along both $\bar{\Gamma}$-$\bar{M}$ (Fig. 3c) and $\bar{\Gamma}$-$\bar{K}$ (Fig. 3e) directions, forming a type-II Dirac nodal loop. The NL2 point is formed by the crossing of the $\gamma_2$ band and the kagome flat band, as shown by the blue and red dashed lines in Fig. 3d, e. Since the kagome flat band is nearly dispersionless, the NL2 Dirac nodal loops and lines can be categorized into type-III.

We have identified the existence of Dirac nodal lines and their momentum distribution in CsTi$_3$Bi$_5$ from our detailed band structure calculations (Fig. 3a, b and Supplementary Fig. 5). We used laser-ARPES to measure CsTi$_3$Bi$_5$ by taking the advantage of its high instrumental resolution. All the measured electronic structures are in a good agreement with the band structure calculations. Specifically, the measured results of the nodal loops at the k$_z$ from our laser-ARPES measurements (Fig. 3c–e) are also consistent with the calculated results (Fig. 3f–h). Our band structure calculations and the laser-ARPES measurements have provided major information on the nodal loops in CsTi$_3$Bi$_5$. To fully characterize the nodal lines in CsTi$_3$Bi$_5$, further ARPES work are needed to probe different k$_z$s by using different photon energies.

## Nontrivial topological surface states

The spin-orbit coupling is stronger in CsTi$_3$Bi$_5$ than that in CsV$_3$Sb$_5$ because of the heavy element Bi. We also note that the calculated energy bands give rise to a strong topological $\mathbb{Z}_2$ index in CsTi$_3$Bi$_5$ (Supplementary Fig. 6)[43,44]. This will result in possible topologically nontrivial surface states. Figure 4a shows the band structure measured around $\bar{\Gamma}$ along the $\bar{M}$-$\bar{\Gamma}$-$\bar{M}$ direction under the LV light polarization. The corresponding second derivative image is shown in Fig. 4b. For comparison, Fig. 4c, d show the calculated band structures without and with SOC, respectively, along the same momentum cut. All the observed bands in Fig. 4b can be well assigned by comparing with the calculated bands (as shown by coloured lines in Fig. 4b, d) except for

one band that is marked as TSS in Fig. 4b. In order to understand its origin, we analyzed the energy bands in details. We found that CsTi$_3$Bi$_5$ has symmetry-protected band degeneracy along the $\Gamma$-A path between the $\gamma$ and $\beta$ bands, as well as between the $\beta$ and $\alpha$ bands, giving rise to multiple topological Dirac semimetal states (Supplementary Fig. 6). This type of topological Dirac semimetal states also appears in AV$_3$Sb$_5$ near the Fermi level along the $\Gamma$-A path. However, in CsTi$_3$Bi$_5$, continuous band gaps throughout the whole Brillouin zone exist between the $\epsilon$ and $\delta$ bands, as well as between the $\delta$ and $\gamma$ bands. Combining the time-reversal and inversion symmetries in CsTi$_3$Bi$_5$, we obtained nontrivial $\mathbb{Z}_2$ topological invariant of $\epsilon$ and $\delta$ bands by calculating the parity of the wavefunctions at all time-reversal invariant momenta (TRIM)[58], as seen in Supplementary Fig. 6. Moreover, band gaps and band inversions due to the strong SOC of the system can induce additional Dirac topological surface states (TSS) crossing at the TRIM $\Gamma$ point as seen in the surface spectral function of Fig. 4e, f. Comparing with the bands in Fig. 4d, we can identify that the topological surface states are located between the $\epsilon$ and $\delta$ bands, which indicates that they are topologically protected by the nontrivial $\mathbb{Z}_2$ invariant. This TSS band gives a good match with the unassigned band in Fig. 4b. Therefore, we provide a definitive spectroscopic evidence that nontrivial topological surface states exist in the kagome superconductor CsTi$_3$Bi$_5$.

## Discussion

In the kagome materials, characteristic electronic structures are predicted that include the flat band across the whole Brillouin zone, van Hove singularities and Dirac points. CsTi$_3$Bi$_5$, as a newly synthesized kagome compound which is isostructural to the AV$_3$Sb$_5$ family, provides a good reference material to fully understand the physics of kagome lattice. We have clearly observed the kagome lattice derived flat band in CsTi$_3$Bi$_5$. Although flat bands are predicted in AV$_3$Sb$_5$, it is not clearly observed in ARPES measurements because one flat band is ~1 eV above the Fermi level that cannot be seen by ARPES while the

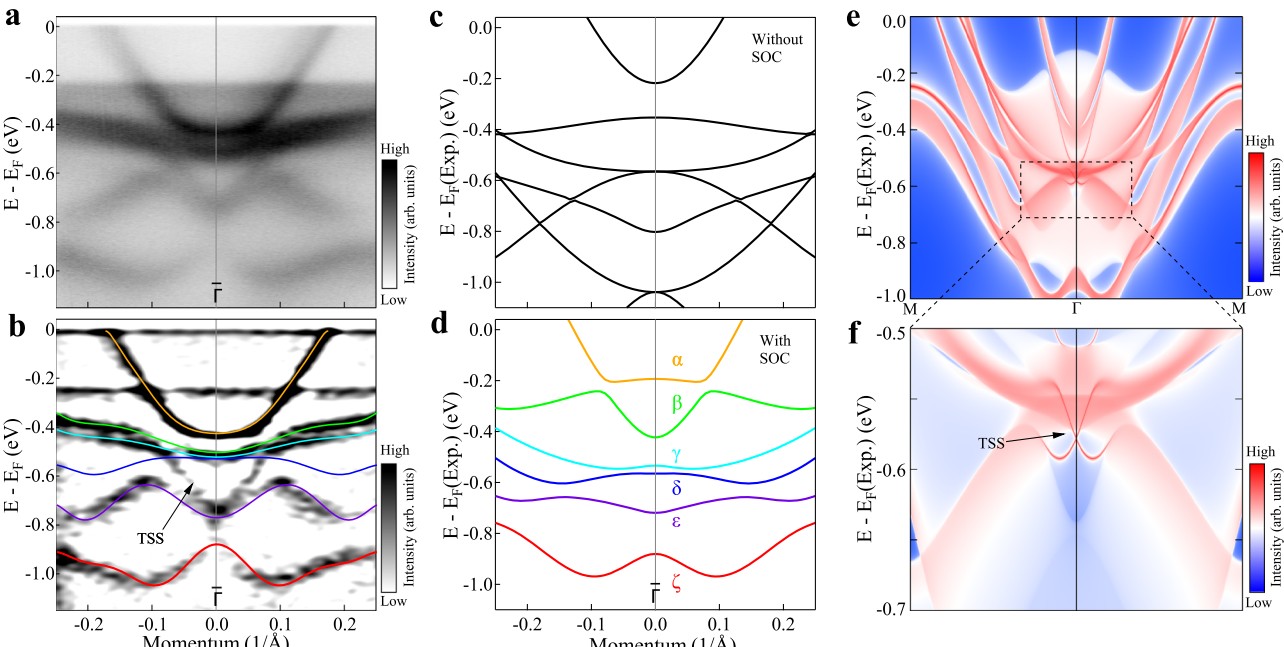

**Fig. 4 | Observation of topological surface states in CsTi$_3$Bi$_5$. a** Band structure measured around $\bar{\Gamma}$ along the $\bar{M}$-$\bar{\Gamma}$-$\bar{M}$ direction under LV polarization geometry. **b** Second derivative image of (**a**). The observed six bands are highlighted by different colored lines. The observed topological surface state (TSS) is marked by an arrow. **c**, **d** The corresponding calculated band structures without (**c**) and with (**d**) SOC. The topological surface state TSS emerges between the $\delta$ and $\epsilon$ bands which

arise from band inversion due to SOC. **e** Calculated surface spectral function along $\bar{M}$-$\bar{\Gamma}$-$\bar{M}$ paths projected on the (001) plane for the Bi-terminated CsTi$_3$Bi$_5$. **f** Enlarged view of the topological surface states (TSS) in (**e**). To make a better comparison with measured results, the calculated Fermi level referred to as E$_F$(Exp.) is shifted downwards by 90 meV.

other flat band is ~1.3 eV below the Fermi level that are not well resolved in ARPES measurements[35,52]. In CsTi$_3$Bi$_5$, it is unambiguous that the expected flat band has been clearly observed that lies close to the Fermi level (~ −0.25 eV). In addition, we also identified type-II and type-III Dirac nodal loops and nodal lines as well as $\mathbb{Z}_2$ nontrivial topological surface states in CsTi$_3$Bi$_5$ that is not observed in AV$_3$Sb$_5$. The spectral weight buildup in the energy range of [−0.25,−0.50] eV across the entire momentum space is also an unexpected observation. Such coexistence of multiple nontrivial band structures in one material provides a platform to explore for novel phenomena and exotic properties that have been expected in the kagome materials such as spin liquid phases[1–4], topological insulator, semimetal and superconductor[5–7], fractional quantum Hall effect[10], quantum anomalous Hall effect[11,12] and unconventional density wave orders[15,16]. So far, many phenomena and physical properties have been observed in AV$_3$Sb$_5$ but their origins are still under investigations[59,60]. Our ARPES investigation on CsTi$_3$Bi$_5$, which is a kagome compound recently synthesized and isostructural to the AV$_3$Sb$_5$, is significant to understand the physics of kagome lattice.

Our present work have demonstrated the tunability of the electronic structures in 135 systems over a large energy range. To realize the expected exotic physical properties in kagome compounds, it is necessary that the related energy scales of the unique electronic structures like the flat band, van Hove singularities and Dirac cones lie close to the Fermi level. In AV$_3$Sb$_5$, the flat bands and the Dirac cones lie far away from the Fermi level while some van Hove singularities are close to the Fermi level[35,36]. Compared with the V 3d orbital derived bands in AV$_3$Sb$_5$, the Ti 3d orbital derived bands are shifted upwards by ~1 eV because of different valance of V and Ti that results in three electrons less per unit cell in CsTi$_3$Bi$_5$ than that in AV$_3$Sb$_5$. Compared with AV$_3$Sb$_5$ where the d$_{x^2−y^2}$/d$_{xy}$-derived flat band is ~1.3 eV below the Fermi level[35], the flat band in CsTi$_3$Bi$_5$ comes much closer ( ~0.25 eV) to the Fermi level. Since this kagome derived flat band lies still away from the Fermi level, it is unlikely that it drives the superconductivity in CsTi$_3$Bi$_5$. Our present work indicates that the position of the flat band, as well as the van Hove singularities, Dirac cones and the nodal loops, can be tuned over a large energy range by manipulating the chemical composition of the 135 kagome compounds. Our results point to a way to move the flat band and other characteristic electronic structures to the Fermi level by further adjusting the composition and doping of the 135 systems that will significantly affect superconductivity and produce exotic physical properties.

In summary, by using high resolution laser based ARPES in combination with the DFT band structure calculations, we have investigated the electronic structure of the newly discovered kagome superconductor CsTi$_3$Bi$_5$. The observed Fermi surface and band structures show excellent agreement with the band structure calculations. We have identified multi-sets of nontrivial band structures in CsTi$_3$Bi$_5$ including the kagome lattice derived flat band, type-II and type-III Dirac nodal loops and nodal lines, as well as $\mathbb{Z}_2$ nontrivial topological surface states. We have demonstrated the tunability of the electronic structures in 135 systems over a large energy range. Such coexistence of tunable nontrivial band structures in one kagome superconductor provides a platform to understand novel phenomena and exotic properties in the kagome materials.

Note added: After the submission of our paper, we became aware of related ARPES work on ATi$_3$Bi$_5$ (A=K, Rb and Cs)[61–63].

## Methods
### Growth of single crystals
CsTi$_3$Bi$_5$ single crystals were grown using a self flux method[44,64]. Typical CsTi$_3$Bi$_5$ crystals with a lateral size of ~3 mm and regular hexagonal morphology were obtained.

### High resolution ARPES measurements
High resolution angle-resolved photoemission measurements were performed using a lab-based ARPES system equipped with the 6.994 eV vacuum-ultra-violet (VUV) laser and a hemispherical electron energy analyzer DA30L (Scienta-Omicron)[65,66]. The laser spot is focused to around 10 um on the sample in order to minimize the influence of sample inhomogeneity. The light polarization can be varied to get linear polarization along different directions. In the LV (LH) polarization the electric vector of the laser light is perpendicular (parallel) to the photoelectron emission plane. The energy resolution was set at 1 meV and the angular resolution was 0.3 degree corresponding to 0.004 $\mathring{A}^{-1}$ momentum resolution at the photon energy of 6.994 eV. All the samples were cleaved in situ at a low temperature of 20 K and measured in ultrahigh vacuum with a base pressure better than 5 x 10$^{-11}$ mbar. The Fermi level is referenced by measuring on clean polycrystalline gold that is electrically connected to the sample.

### Band calculations
Density functional theory (DFT) calculations with projector augmented-wave pseudopotential method[67] are implemented through Vienna ab initio simulation package (VASP)[68]. The exchange correlation functional is treated by Perdew-Burke-Ernzerh (PBE) of parameterization of generalized gradient approximation (GGA)[69]. The convergence criterion of atomic forces in structural optimization with VASP is less than 1 meV/$\mathring{A}$ total energy convergence threshold of all processes is 10$^{-6}$ eV/atom. The cutoff energy of the plane-wave is set as 520 eV. The Γ centered 20 × 20 × 12 Monkhorst-Pack k-points grid is used in the self-consistent cycle. Wannier90 package[70] is used to fit Wannier functions and construct tight-binding models, and WannierTools[71] package is used to calculate the surface spectral functions by using the surface Green's function method. Calculations of structures' parity are performed through a combination of the irvsp program[72] and VASP.

## Data availability
All data are processed by using Igor Pro 8.02 software. All data needed to evaluate the conclusions in the paper are available within the article and its Supplementary Information files. All raw data generated during the current study are available from the corresponding author upon request.

## Code availability
The codes used for the DFT calculations in this study are available from the corresponding authors upon request.

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

## Acknowledgements

We thank X. L. Dong and J. L. Liu for their help in sample characterizations. This work is supported by the National Natural Science Foundation of China (Grant No. 11888101, 11922414, 11974404 and 11834014), the National Key Research and Development Program of China (Grant No. 2021YFA1401800, 2017YFA0302900, 2018YFA0704200, 2018YFA0305600, 2018YFA0305800, 2019YFA0308000 and 2022YFA1604200), the Strategic Priority Research Program (B) of the Chinese Academy of Sciences (Grant No. XDB25000000, XDB28000000 and XDB33000000), the Innovation Program for Quantum Science and Technology (Grant No. 2021ZD0301800), the Youth Innovation Promotion Association of CAS (Grant No. Y2021006) and Synergetic Extreme Condition User Facility (SECUF).

## Author contributions

J.G.Y., X.W.Y., Z.Z., and Y.Y.X. contribute equally to this work. X.J.Z., L.Z., J.G.Y., and Y.Y.X. proposed and designed the research. Z.Z., Y.H.Y., Hu.C., H.T.Y., and H.J.G. contributed to single crystal growth. X.W.Y., J.Y.Y., B.G., and G.S. contribute to the DFT band calculations. T.M.M., H.L.L., Ha.C., B.L., W.P.Z., S.J.Z., F.F.Z., F.Y., Z.M.W., Q.J.P., H.Q.M., G.D.L., L.Z., Z.Y.X. and X.J.Z. contributed to the development and maintenance of Laser-ARPES system. J.G.Y. and Y.Y.X. carried out the ARPES experiment. J.G.Y., L.Z. and X.J.Z. analyzed the data. J.G.Y., L.Z. and X.J.Z. wrote the paper with X.W.Y., H.J.G., and G.S.; All authors participated in the discussion and comment on the paper.

## Competing interests

The authors declare no competing interests.
