## [Peer Review File · Nature Communications]

Observation of Flat Band, Dirac Nodal Lines and Topological Surface States in Kagome Superconductor CsTi₃Bi₅REVIEWER COMMENTS

Reviewer #1 (Remarks to the Author):

This manuscript describes a first-principle calculation and ARPES study of CsTi₃Bi₅, a kagome compound recently synthesized. It is isostructural to the celebrated AV₃Sb₅ family, but with a different effective doping. Therefore, an ARPES investigation is indeed very interesting. The DFT calculation is similar to earlier reports (ref. 44 for example).

The ARPES data presented here are of high quality and they arguably bring information useful for the community. They are compared to DFT calculations to « confirm » the presence of some features, such as flat bands or topological surface states. This is a standard way to proceed, but I find that one does not learn very much on the physics of this material in the process. The fact that the *k_z* dimension cannot be explored by laser ARPES is a shortcoming of this study, rather minor for this 2D material, but more serious to locate precisely the Dirac loops.

The most interesting finding for further studies of this material is probably the Fermi Surface map and the observation of a flat band, relatively close to the Fermi level. However, contrary to what the authors claim many times, they do not demonstrate it is built from destructive interference on the kagome lattice, independently of the suggestion of the calculation. An associated finding is the region of high intensity between -0.2 and -0.5eV (Fig. S1). This is quite intriguing but still lacks a clear explanation.

Although I appreciate the data presentation in general, I think the unsymmetrized FS of Fig. 1 should be given, at least in supplementary information.

To summarize, this study characterizes well the electronic structure of this interesting material, but it is not extremely unexpected, given the previous knowledge on AV₃Sb₅ and the early reports on the synthesis of ATi₃Sb₅. Although it is of good quality, I find it lacks a specific discovery that would clearly justify publication in Nature Communication.

Reviewer #2 (Remarks to the Author):

Yang et al. study the new Kagome superconductor CsTi₃Bi₅ via a combination of density functional theory and angle-resolved photoemission. They discover evidence for flat bands below the Fermi energy, and type-II and type-III Dirac nodes. In addition they report evidence of topological surface states.

This paper is well-written and makes claims that are supported by the evidence. I do not see any issues with this manuscript and I would be happy for it to be published as is. My only question is that the flat bands observed by the authors are still significantly below the Fermi level - buried 200meV below - so it does not seem likely that this band drives the superconductivity. Perhaps the authors could comment on this further.

Reviewer #3 (Remarks to the Author):

The manuscript by Yang et al. presents laser-based angle-resolved photoemission spectroscopy (ARPES) measurements, supplemented by density functional theory (DFT) calculations, of a lesser-studied Kagome superconductor CsTi₃Bi₅. This material is isostructural to its celebrated cousin

AV3Sb5 (A = Rb, K, Cs), but a different band filling shifts the positions of the interesting points in the band structure with respect to the Fermi level. The reported ARPES measurements are of high quality and showcase interesting features in the band structure, but the discussion of their relevance to the physics of Kagome metals is rather limited. The manuscript is written in a nice style carefully introducing all the findings. At the present stage, however, I am not fully convinced of its suitability for Nature Communications.

I have following criticisms and comments regarding the manuscript:

- The discussion of nodal loops relies only on the DFT calculations. It would be very helpful if the authors could present synchrotron-based photon energy-variable data to confirm this point experimentally.
- I have some doubts regarding the flat bands shown in Fig. 2. There is a cutoff of background intensity around the binding energy where the theoretically predicted flat bands are, and this shows in the 2nd derivative plots as an extended flat band. The authors should explain the origin of this "spectral weight buildup" and why they consider its cutoff to be a real part of the band structure and not merely an artifact.
- The same consideration applies to the type-III Weyl points. The non-dispersing branch is the cutoff of the spectral weight buildup.
- The salient features of the band structure discussed in this manuscript are placed at least 250 meV away from the Fermi level. The authors should comment on the possible role they could play in the physics of Kagome metals.

Response to Reviewer's Comments

Response to Reviewer #1's comments

This manuscript describes a first-principle calculation and ARPES study of CsTi₃Bi₅, a kagome compound recently synthesized. It is isostructural to the celebrated AV₃Sb₅ family, but with a different effective doping. Therefore, an ARPES investigation is indeed very interesting. The DFT calculation is similar to earlier reports (ref. 44 for example).

We thank Reviewer #1 for the careful reviewing of our paper and providing constructive comments and suggestions to improve our paper. We thank the Reviewer for pointing out that an ARPES investigation of CsTi₃Bi₅ is indeed very interesting.

The ARPES data presented here are of high quality and they arguably bring information useful for the community. They are compared to DFT calculations to « confirm » the presence of some features, such as flat bands or topological surface states. This is a standard way to proceed, but I find that one does not learn very much on the physics of this material in the process. The fact that the kz dimension cannot be explored by laser ARPES is a shortcoming of this study, rather minor for this 2D material, but more serious to locate precisely the Dirac loops.

We thank the Reviewer for capturing the high quality of our data and significance of our work. In the study of topological materials, it is a standard way to first have band structure predictions and then experimental (mostly ARPES) confirmations. For materials with relatively weak electron correlation, the agreement between band structure calculations and experiments is usually high. Although band structure calculations can do a good job, the actual ARPES measurements are necessary to pin down on the electronic structure and check on the band structure calculations. Our present ARPES study of CsTi₃Bi₅ is interesting and significant because of the following reasons. First, CsTi₃Bi₅ is a new kagome compound recently synthesized which is isostructural to the celebrated AV₃Sb₅ family but with a different effective doping. Therefore, an ARPES investigation is very interesting to provide more complete information to understand the physics of Kagome lattice. Second, although CsTi₃Bi₅ is isostructural to the AV₃Sb₅ family, the physical properties and related physics can be rather different because the relative chemical potential shift reaches up to ~1eV so the energy scales of the flat band, van Hove singularities and Dirac cones are quite different. Third, although flat bands are predicted in AV₃Sb₅, it is not clearly observed in ARPES measurements because one flat band is ~1eV above the Fermi level that cannot be seen by ARPES while the other flat band is ~1.3eV below the Fermi level that are not well resolved in ARPES measurements (M. Kang et al., Nat. Phys. 18, 301 (2022) and Y. Hu et al., Science Bulletin 67, 495 (2022)). In CsTi₃Bi₅, it is the first time that the

expected flat band has been clearly observed that lies close to the Fermi level (-0.25eV). Fourth, the topological surface state we observed in CsTi₃Bi₅ is not observed in AV₃Sb₅. Therefore, CsTi₃Bi₅ provides a new platform to study the kagome lattice related physics and phenomena with reference to AV₃Sb₅. The simultaneous existence of multi-sets of nontrivial band structures in one kagome superconductor, in particular the observation of the flat band, is significant to study related physics and realize novel quantum phenomena in the kagome lattice.

As the Reviewer points out, since CsTi₃Bi₅ is a 2D material, our laser-ARPES measurements are sufficient to study the major electronic structure of the material. In the present paper, we used laser-ARPES to measure CsTi₃Bi₅ by taking the advantage of high instrumental resolution. Overall the measured electronic structures are in a good agreement with the band structure calculations. The measured results of the nodal loops at the kz from our laser-ARPES are also consistent with the calculated results. Our laser-ARPES measurement and the band structure calculations have provided major information on the nodal loops in CsTi₃Bi₅ although we agree with the Reviewer that their complete characterization needs future work using different photon energies.

Following the Reviewer's comment, in the revised manuscript, we added discussions on the novelty and significance of our present work.

The most interesting finding for further studies of this material is probably the Fermi Surface map and the observation of a flat band, relatively close to the Fermi level. However, contrary to what the authors claim many times, they do not demonstrate it is built from destructive interference on the kagome lattice, independently of the suggestion of the calculation. An associated finding is the region of high intensity between -0.2 and -0.5eV (Fig. S1). This is quite intriguing but still lacks a clear explanation.

The band structure measurements from ARPES can not directly tell whether or not the observed flat band is from destructive interference on the kagome lattice. To understand the origin of the flat band in CsTi₃Bi₅, it is necessary to rely on band structure calculations. To further understand the flat band in CsTi₃Bi₅, we carried out band structure calculations to get the charge density distributions (Fig. R1f below). These features are consistent with those of the flat band of the tight-binding model in Fig. R1(a-c). These results show that the flat band formed by Ti-dx²-y²/dxy orbital near -0.25 eV originates from the destructive interference mechanism of the kagome lattice.

Fig. R1 Origin of the flat band in CsTi₃Bi₅. (a-c) Electronic band structures of the tight-binding model of kagome lattice weighted by projected composition of three sublattices A, B, and C, respectively. (d) The localized eigenstate of the flat band in (a-c), where “+” and “-” represent the sign of wavefunctions of different sites. (e) DFT electronic band structures of CsTi₃Bi₅ with considering SOC. (f) Charge density distribution of the flat band states at the M, M’, and K respectively. The charge density distribution is drawn with an isosurface of about 0.004 e/Bohr³. These three electronic states are marked by blue arrows in (e).

Following the Reviewer’s comment, we added Fig. R1 as a new Fig. S2 and discussions on the origin of the flat band in Supplementary Materials.

Following the Reviewer’s comment, we also added Fig. R2 below as Fig. S4 in Supplementary Materials and added more discussions in the revised manuscript on the intensity buildup: “The observation of high intensity buildup between [-0.25,-0.50] eV in CsTi₃Bi₅ is an unexpected new finding. It is not expected from the band structure calculations. To the best of our knowledge, such a spectral buildup over the entire Brillouin zone has not been observed by ARPES in other materials. One possibility to check is whether this may be attributed to the *k_z* effect. Due to finite *k_z* resolution, the measured band structure may correspond to the summation of bands at different *k_z*s. The measured data are found to be not consistent with the calculated results (*k_z* = 0~1), indicating that the *k_z* effect is unlikely (Fig. S4). The two cutoff energies of the spectral weight buildup happen to coincide with the top and bottom energy positions of the kagome derived flat band. This suggests that the extra spectral weight buildup is closely related to the existence of the flat band. It is possible that the spectral buildup may come from electron scattering of the flat band states. Since the flat band corresponds to high density of states confined by the band top (~-0.25eV) and the band bottom (~-0.5eV), the scattered electrons may lie in the same energy range. At present, we can not fully pin down the origin of the high intensity buildup whether it is an extrinsic effect or it represents an intrinsic effect due to other origins. Further efforts are needed to fully understand this interesting phenomenon.”

Fig. R2 Comparison between the measured band structures and the calculated k_z -integrated band structures for CsTi_3Bi_5 . (a,b) Detailed band structures measured along the $\bar{\Gamma}-\bar{M}$ and $\bar{\Gamma}-\bar{K}$ high symmetry directions, respectively. The spectral weight buildup regions between $-0.25\sim-0.50$ eV are marked by the dashed red frames. (c,d) Calculated k_z -integrated band structures along the $\bar{\Gamma}-\bar{M}$ and $\bar{\Gamma}-\bar{K}$ high symmetry directions, respectively.

Although I appreciate the data presentation in general, I think the unsymmetrized FS of Fig. 1 should be given, at least in supplementary information.

Following Reviewer #1's suggestion, we added the unsymmetrized Fermi surfaces of Fig. 1 in Fig. S1 in the revised Supplementary Materials (also shown in Fig. R3 below).

Fig. R3 Original Fermi surface mappings of CsTi_3Bi_5 over three momentum regions. The Fermi surface in Fig. 1 is obtained from these results by considering six-fold symmetry.

To summarize, this study characterizes well the electronic structure of this interesting material, but it is not extremely unexpected, given the previous knowledge on AV₃Sb₅ and the early reports on the synthesis of ATi₃Sb₅. Although it is of good quality, I find it lacks a specific discovery that would clearly justify publication in Nature Communication.

We thank the Reviewer in pointing out that our study characterizes well the electronic structure and the studied material is interesting. We also thank the Reviewer in pointing out that our paper is of good quality.

For materials with relatively weak electron correlation, modern band structure calculations can well capture the major electronic structures. This is why the measured ARPES results of AV₃Sb₅ and ATi₃Bi₅ are not extremely unexpected when compared with the band structure calculations. Although band structure calculations can do a good job, the actual ARPES measurements are necessary to pin down on the electronic structure and check on the band structure calculations. Our present ARPES study of CsTi₃Bi₅ is interesting and significant because of the following reasons. First, as the Reviewer pointed out, CsTi₃Bi₅ is a new kagome compound recently synthesized which is isostructural to the celebrated AV₃Sb₅ family but with a different effective doping. Therefore, an ARPES investigation is indeed very interesting to provide more complete information to understand the physics of Kagome lattice. Second, although CsTi₃Bi₅ is isostructural to the AV₃Sb₅ family, the physical properties and related physics can be rather different because the relative chemical potential shift reaches up to ~1eV so the energy scales of the flat band, van Hove singularities and Dirac cones are quite different. Third, although flat bands are predicted in AV₃Sb₅, it is not clearly observed in ARPES measurements because one flat band is ~1eV above the Fermi level that cannot be seen by ARPES while the other flat band is ~1.3eV below the Fermi level that are not well resolved in ARPES measurements (M. Kang et al., Nat. Phys. 18, 301 (2022) and Y. Hu et al., Science Bulletin 67, 495 (2022)). In CsTi₃Bi₅, it is the first time that the expected flat band has been clearly observed that lies close to the Fermi level (-0.25eV). Fourth, the topological surface state we observed in CsTi₃Bi₅ is not observed in AV₃Sb₅. Therefore, CsTi₃Bi₅ provides a new platform to study the kagome lattice related physics and phenomena with reference to AV₃Sb₅. The simultaneous existence of multi-sets of nontrivial band structures in one kagome superconductor, in particular the observation of the flat band, is significant to study related physics and realize novel quantum phenomena in the kagome lattice.

Response to Reviewer #2's comments

Yang et al. study the new Kagome superconductor CsTi₃Bi₅ via a combination of density functional theory and angle-resolved photoemission. The discover evidence for flat bands below the Fermi energy, and type-II and type-III Dirac nodes. In addition they report evidence of topological surface states.

This paper is well-written and makes claims that are supported by the evidence. I do not see any issues with this manuscript and I would be happy for it to be published as is. My only question is that the flat bands observed by the authors are still significantly below the Fermi level - buried 200meV below - so it does not seem likely that this band drives the superconductivity. Perhaps the authors could comment on this further.

We thank the Reviewer for the careful reviewing of our paper and providing constructive comments and suggestions to improve our paper. The Reviewer nicely captured the main results of our paper. We also thank the Reviewer for pointing out that this paper is well-written and makes claims that are supported by the evidence and recommending it to be published as is.

We agree with the Reviewer that, since the flat band lies $\sim 250\text{meV}$ below the Fermi level, it is unlikely that it drives the superconductivity in CsTi_3Bi_5 . But the clear observation of the flat band in CsTi_3Bi_5 is significant. First, compared with AV_3Sb_5 where the flat bands are $\sim 1\text{eV}$ away from the Fermi level, the flat band in CsTi_3Bi_5 comes much closer ($\sim 0.25\text{eV}$) to the Fermi level. Second, compared with AV_3Sb_5 , the relative energy position of the flat band is shifted upwards by $\sim 1\text{eV}$. This indicates that the position of the flat band can be tuned over a large energy range by manipulating the chemical composition of the 135 kagome compounds. Our results point to a way to move the flat band near the Fermi level by further adjusting the composition and doping of the 135 systems that will significantly affect superconductivity and produce exotic physical properties.

Following the Reviewer's comment, we added the above discussions in the revised manuscript "Since this kagome derived flat band lies still away from the Fermi level, it is unlikely that it drives the superconductivity in CsTi_3Bi_5 . Our present work indicates that the position of the flat band, as well as the van Hove singularities, Dirac cones and the nodal loops, can be tuned over a large energy range by manipulating the chemical composition of the 135 kagome compounds. Our results point to a way to move the flat band and other characteristic electronic structures to the Fermi level by further adjusting the composition and doping of the 135 systems that will significantly affect superconductivity and produce exotic physical properties."

Response to Reviewer #3's comments

The manuscript by Yang et al. presents laser-based angle-resolved photoemission spectroscopy (ARPES) measurements, supplemented by density functional theory (DFT) calculations, of a lesser-studied Kagome superconductor CsTi_3Bi_5 . This material is isostructural to its celebrated cousin AV_3Sb_5 ($A = \text{Rb}, \text{K}, \text{Cs}$), but a different band filling shifts the positions of the interesting points in the band structure with respect to the Fermi level. The reported ARPES measurements are of high quality and showcase interesting features in the band structure, but the discussion of

their relevance to the physics of Kagome metals is rather limited. The manuscript is written in a nice style carefully introducing all the findings. At the present stage, however, I am not fully convinced of its suitability for Nature Communications.

We thank the Reviewer for the careful reviewing of our paper and providing constructive comments and suggestions to improve our paper. The Reviewer nicely captured the main results of our paper. We also thank the Reviewer for pointing out that our reported ARPES measurements are of high quality and showcase interesting features in the band structure and that the manuscript is written in a nice style carefully introducing all the findings.

Following the Reviewer's comments, in the revised manuscript, we added discussions on the relevance of our results to the physics of the Kagome metal "In the Kagome materials, characteristic electronic structures are predicted that include the flat band across the whole Brillouin zone, van Hove singularities and Dirac points. CsTi₃Bi₅, as a newly synthesized kagome compound which is isostructural to the AV₃Sb₅ family, provides a good reference material to fully understand the physics of Kagome lattice. We have clearly observed the kagome lattice derived flat band in CsTi₃Bi₅. Although flat bands are predicted in AV₃Sb₅, it is not clearly observed in ARPES measurements because one flat band is ~1 eV above the Fermi level that cannot be seen by ARPES while the other flat band is ~1.3 eV below the Fermi level that are not well resolved in ARPES measurements[35, 61]. In CsTi₃Bi₅, it is the first time that the expected flat band has been clearly observed that lies close to the Fermi level (~-0.25 eV). In addition, we also identified type-II and type-III Dirac nodal loops and nodal lines as well as Z₂ nontrivial topological surface states in CsTi₃Bi₅ that is not observed in AV₃Sb₅. The spectral weight buildup in the energy range of [-0.25,-0.50] eV across the entire momentum space is also a new observation. Such coexistence of multiple nontrivial band structures in one material provides a new platform to explore for novel phenomena and exotic properties that have been expected in the kagome materials such as spin liquid phases[1–4], topological insulator, semimetal and superconductor[5–7], fractional quantum Hall effect[10], quantum anomalous Hall effect[11, 12] and unconventional density wave orders[15, 16]. So far, many new phenomena and physical properties have been observed in AV₃Sb₅ but their origins are still under investigations[68, 69]. Our ARPES investigation on CsTi₃Bi₅, which is a new Kagome compound recently synthesized and isostructural to the AV₃Sb₅, is significant to understand the physics of Kagome lattice.

Our present work have demonstrated the tunability of the electronic structures in 135 systems over a large energy range. To realize the expected exotic physical properties in kagome compounds, it is necessary that the related energy scales of the unique electronic structures like the flat band, van Hove singularities and Dirac cones lie close to the Fermi level. In AV₃Sb₅, the flat bands and the Dirac cones lie far away from the Fermi level while some van Hove singularities are close to the Fermi level[35, 36]. Compared with the V 3d orbital derived bands in AV₃Sb₅, the Ti 3d orbital derived bands are shifted upwards by ~1 eV because of different valance of V and Ti that results

in three electrons less per unit cell in CsTi₃Bi₅ than that in AV₃Sb₅. Compared with AV₃Sb₅ where the $d_{x^2-y^2}/d_{xy}$ -derived flat band is ~ 1.3 eV below the Fermi level[35], the flat band in CsTi₃Bi₅ comes much closer (~ 0.25 eV) to the Fermi level. Since this kagome derived flat band lies still away from the Fermi level, it is unlikely that it drives the superconductivity in CsTi₃Bi₅. Our present work indicates that the position of the flat band, as well as the van Hove singularities, Dirac cones and the nodal loops, can be tuned over a large energy range by manipulating the chemical composition of the kagome compounds. Our results point to a way to move the flat band and other characteristic electronic structures to the Fermi level by further adjusting the composition and doping of the 135 systems that will significantly affect superconductivity and produce exotic physical properties.”

The discussion of nodal loops relies only on the DFT calculations. It would be very helpful if the authors could present synchrotron-based photon energy-variable data to confirm this point experimentally.

In the present paper, we used laser-ARPES to measure CsTi₃Bi₅ by taking the advantage of high instrumental resolution. Overall the measured electronic structures are in a good agreement with the band structure calculations. The measured results of the nodal loops at the k_z from our laser-ARPES are also consistent with the calculated results. Our laser-ARPES measurement and the band structure calculations have provided major information on the nodal loops in CsTi₃Bi₅ although we agree with the Reviewer that their complete characterization needs future work using different photon energies from synchrotron radiation light sources.

I have some doubts regarding the flat bands shown in Fig. 2. There is a cutoff of background intensity around the binding energy where the theoretically predicted flat bands are, and this shows in the 2nd derivative plots as an extended flat band. The authors should explain the origin of this “spectral weight buildup” and why they consider its cutoff to be a real part of the band structure and not merely an artifact.

We observed the spectral weight buildup between $-0.25 \sim -0.50$ eV energy range across the entire Brillouin zone as shown in Fig. 2(a,b,c) and Fig. S3. This spectral weight buildup is bounded by two cutoffs: the upper bound at -0.25 eV and the lower bound at -0.50 eV. The second derivative (Fig. 2c) makes the two boundaries show up like two bands but they are not the real part of the band structure, as we pointed out in the paper that “The second part flat band at ~ 250 meV actually represents a spectral weight cut-off at this energy (see Fig. S3 in Supplementary Materials for details).”

The observation of high intensity buildup between $E_B = 0.25 \sim 0.5$ eV in CsTi₃Bi₅ is an unexpected new finding. It is not expected from the band structure calculations. To the best of our knowledge, such a spectral buildup over the entire BZ has not been observed

by ARPES in other materials. One possibility we thought about is whether this may be attributed to the k_z effect. Due to finite k_z resolution, the measured band structure may correspond to the summation of bands at different k_z s. Careful comparison between the measured results and calculated results ($k_z = 0 \sim 1$) (Fig. R2) indicates that it is unlikely. The second possibility to consider is whether the spectral buildup is from electron scattering of the flat band states. Since the flat band corresponds to high density of states confined by the band top (~ -0.25 eV) and the band bottom (~ -0.5 eV), the scattered electrons may lie in this energy range. At present, we do not know clearly the origin of the high intensity buildup and further efforts are needed to fully understand this interesting phenomenon.

To clarify the nature of the spectral weight cutoffs, in the revised manuscript, we added “This spectral weight buildup is bounded by two cutoffs: the upper bound at -0.25 eV and the lower bound at -0.50 eV. The second derivative (Fig. 2c) makes the two boundaries show up like two bands but they are not the real part of the band structure”. Following the Reviewer’s comment, we also added discussions on the origin of the spectral weight buildup: “The observation of high intensity buildup between $[-0.25, -0.50]$ eV in CsTi_3Bi_5 is an unexpected new finding. It is not expected from the band structure calculations. To the best of our knowledge, such a spectral buildup over the entire Brillouin zone has not been observed by ARPES in other materials. One possibility to check is whether this may be attributed to the k_z effect. Due to finite k_z resolution, the measured band structure may correspond to the summation of bands at different k_z s. The measured data are found to be not consistent with the calculated results ($k_z = 0 \sim 1$), indicating that the k_z effect is unlikely (Fig. S4). The two cutoff energies of the spectral weight buildup happen to coincide with the top and bottom energy positions of the kagome derived flat band. This suggests that the extra spectral weight buildup is closely related to the existence of the flat band. It is possible that the spectral buildup may come from electron scattering of the flat band states. Since the flat band corresponds to high density of states confined by the band top (~ -0.25 eV) and the band bottom (~ -0.5 eV), the scattered electrons may lie in the same energy range. At present, we cannot fully pin down the origin of the high intensity buildup whether it is an extrinsic effect or it represents an intrinsic effect due to other origins. Further efforts are needed to fully understand this interesting phenomenon.”

The same consideration applies to the type-III Weyl points. The non-dispersing branch is the cutoff of the spectral weight buildup.

The non-dispersive branch at ~ -0.25 eV in Fig. 3c is from the cutoff of the spectral weight buildup. The features at ~ -0.25 eV in Fig. 3d and 3e represent real bands as can be seen and analyzed from the original data (Fig. 2a and 2b). They are also consistent with the band structure calculations (Fig. 3g and 3h).

Following the Reviewer’s comment, in the revised manuscript, we added in the figure caption of Fig. 3 “We note that the non-dispersive branch at ~ -0.25 eV in c is from the

cutoff of the spectral weight buildup. The features at ~ -0.25 eV in d and e represent real bands as can be seen and analyzed from the original data (Fig. 2a and 2b).”

The salient features of the band structure discussed in this manuscript are placed at least 250 meV away from the Fermi level. The authors should comment on the possible role they could play in the physics of Kagome metals.

Following the Reviewer’s suggestion, we added related discussions in the revised manuscript “Our present work have demonstrated the tunability of the electronic structures in 135 systems over a large energy range. To realize the expected exotic physical properties in kagome compounds, it is necessary that the related energy scales of the unique electronic structures like the flat band, van Hove singularities and Dirac cones lie close to the Fermi level. In AV_3Sb_5 , the flat bands and the Dirac cones lie far away from the Fermi level while some van Hove singularities are close to the Fermi level[35, 36]. Compared with the V 3d orbital derived bands in AV_3Sb_5 , the Ti 3d orbital derived bands are shifted upwards by ~ 1 eV because of different valance of V and Ti that results in three electrons less per unit cell in $CsTi_3Bi_5$ than that in AV_3Sb_5 . Compared with AV_3Sb_5 where the dx^2-y^2/dxy -derived flat band is ~ 1.3 eV below the Fermi level[35], the flat band in $CsTi_3Bi_5$ comes much closer (~ 0.25 eV) to the Fermi level. Since this kagome derived flat band lies still away from the Fermi level, it is unlikely that it drives the superconductivity in $CsTi_3Bi_5$. Our present work indicates that the position of the flat band, as well as the van Hove singularities, Dirac cones and the nodal loops, can be tuned over a large energy range by manipulating the chemical composition of the kagome compounds. Our results point to a way to move the flat band and other characteristic electronic structures to the Fermi level by further adjusting the composition and doping of the 135 systems that will significantly affect superconductivity and produce exotic physical properties.”

Summary of Modifications

1. Following Reviewer #1's suggestion, we added the unsymmetrized FS of Fig. 1 in Fig. S1 in the revised Supplementary Materials.
2. Following Reviewer #1's suggestion, we added more discussions (Fig. S2) about the "destructive interference" in revised Supplementary Materials.
3. Following the Reviewer #1 and #3's suggestions, on page 7, line 169, we added more discussion about the spectral buildup "The observation of high intensity buildup between [-0.25, -0.50] eV in CsTi₃Bi₅ is an unexpected new finding. It is not expected from the band structure calculations. To the best of our knowledge, such a spectral buildup over the entire Brillouin zone has not been observed by ARPES in other materials. One possibility to check is whether this may be attributed to the k_z effect. Due to finite k_z resolution, the measured band structure may correspond to the summation of bands at different k_z s. The measured data are found to be not consistent with the calculated results ($k_z = 0\sim 1$), indicating that the k_z effect is unlikely (Fig. S4). The two cutoff energies of the spectral weight buildup happen to coincide with the top and bottom energy positions of the kagome derived flat band. This suggests that the extra spectral weight buildup is closely related to the existence of the flat band. It is possible that the spectral buildup may come from electron scattering of the flat band states. Since the flat band corresponds to high density of states confined by the band top (~ -0.25 eV) and the band bottom (~ -0.5 eV), the scattered electrons may lie in the same energy range. At present, we can not fully pin down the origin of the high intensity buildup whether it is an extrinsic effect or it represents an intrinsic effect due to other origins. Further efforts are needed to fully understand this interesting phenomenon."
4. A new Fig. S4 is added in revised Supplementary Materials.
5. Following the Reviewer #2 and #3's suggestions, on page 9, line 253, we added discussions about the main findings and the significance of our work "In the Kagome materials, characteristic electronic structures are predicted that include the flat band across the whole Brillouin zone, van Hove singularities and Dirac points. CsTi₃Bi₅, as a newly synthesized kagome compound which is isostructural to the AV₃Sb₅ family, provides a good reference material to fully understand the physics of Kagome lattice. We have clearly observed the kagome lattice derived flat band in CsTi₃Bi₅. Although flat bands are predicted in AV₃Sb₅, it is not clearly observed in ARPES measurements because one flat band is ~ 1 eV above the Fermi level that cannot be seen by ARPES while the other flat band is ~ 1.3 eV below the Fermi level that are not well resolved in ARPES measurements[35, 61]. In CsTi₃Bi₅, it is the first time that the expected flat band has been clearly observed that lies close to the Fermi level (~ -0.25 eV). In addition, we also identified type-II and type-III Dirac nodal loops and nodal lines as well as Z₂ nontrivial

topological surface states in CsTi₃Bi₅ that is not observed in AV₃Sb₅. The spectral weight buildup in the energy range of [-0.25,-0.50] eV across the entire momentum space is also a new observation. Such coexistence of multiple nontrivial band structures in one material provides a new platform to explore for novel phenomena and exotic properties that have been expected in the kagome materials such as spin liquid phases[1–4], topological insulator, semimetal and superconductor[5–7], fractional quantum Hall effect[10], quantum anomalous Hall effect[11, 12] and unconventional density wave orders[15, 16]. So far, many new phenomena and physical properties have been observed in AV₃Sb₅ but their origins are still under investigations[68, 69]. Our ARPES investigation on CsTi₃Bi₅, which is a new Kagome compound recently synthesized and isostructural to the AV₃Sb₅, is significant to understand the physics of Kagome lattice.

Our present work have demonstrated the tunability of the electronic structures in 135 systems over a large energy range. To realize the expected exotic physical properties in kagome compounds, it is necessary that the related energy scales of the unique electronic structures like the flat band, van Hove singularities and Dirac cones lie close to the Fermi level. In AV₃Sb₅, the flat bands and the Dirac cones lie far away from the Fermi level while some van Hove singularities are close to the Fermi level[35, 36]. Compared with the V 3d orbital derived bands in AV₃Sb₅, the Ti 3d orbital derived bands are shifted upwards by ~1 eV because of different valance of V and Ti that results in three electrons less per unit cell in CsTi₃Bi₅ than that in AV₃Sb₅. Compared with AV₃Sb₅ where the d_{x²-y²}/d_{xy}-derived flat band is ~1.3 eV below the Fermi level[35], the flat band in CsTi₃Bi₅ comes much closer (~0.25 eV) to the Fermi level. Since this kagome derived flat band lies still away from the Fermi level, it is unlikely that it drives the superconductivity in CsTi₃Bi₅. Our present work indicates that the position of the flat band, as well as the van Hove singularities, Dirac cones and the nodal loops, can be tuned over a large energy range by manipulating the chemical composition of the kagome compounds. Our results point to a way to move the flat band and other characteristic electronic structures to the Fermi level by further adjusting the composition and doping of the 135 systems that will significantly affect superconductivity and produce exotic physical properties.”

6. On page 6, line 163, we changed “The two energies happen to coincide with the top and bottom energy positions of the first part flat band. This suggests that the extra spectral weight buildup is closely related to the first part flat band. The second part flat band at ~250 meV actually represents a spectral weight cut-off at this energy (see Fig. S1 in Supplementary Materials for details).” into “This spectral weight buildup is bounded by two cutoffs: the upper bound at -0.25 eV and the lower bound at -0.50 eV. The second derivative (Fig. 2c) makes the two boundaries show up like two bands but they are not the real part of the band 166 structure. Therefore, the second part flat band at ~250 meV actually represents a spectral weight cutoff at this energy (see Fig. S3 in Supplementary Materials for details).”

7. Following the Reviewer #3's comment, on page 22, in the figure caption of Fig. 3, we added "We note that the non-dispersive branch at ~ -0.25 eV in c is from the cutoff of the spectral weight buildup. The features at ~ -0.25 eV in d and e represent real bands as can be seen and analyzed from the original data (Fig. 2a and 2b)."
8. We replaced Figure 4e and 4f with correct energy shift.

REVIEWER COMMENTS

Reviewer #1 (Remarks to the Author):

The 3 referees seem to give a similar appreciation of the manuscript : it is well written and gives a high quality ARPES view of the electronic structure of this interesting kagome material, but it fails a bit short of giving a really unexpected and significant progress into the physics of these materials. In their answer, the authors can only repeat why they think kagome physics is interesting in general. I largely agree with them, but I still think that a paper in Nature Communications should go a bit beyond this. I would rather recommend this manuscript for a more specialized journal.

I appreciate the addition of unsymmetrized FS and sublattice decomposition of the band structure, in the supplementary data of the revised manuscript.

Reviewer #3 (Remarks to the Author):

The changes to the manuscript have made the motivation of the study stronger, but overall there is still no clear sense of why any of the features are important to the material's properties or what is the novelty on a technical or conceptual level. I do not find the other revisions substantial enough to recommend the publication.

In particular, while I agree that the band structure calculation is expected to give a faithful description of a relatively low correlation material, I think the lack of k_z -dependent measurement is a major shortcoming when discussing the discovery of nodal loops in an experimental paper. Any statements in this regard should be softened and should not be presented as a major point in the study, which considerably weakens the overall message.

With that said, I would like to point out that the work is clearly interesting to the condensed matter community as evidenced by several manuscripts which recently appeared on arXiv (while I do understand that the authors had no knowledge of them at the time of original submission, I believe these should also be acknowledged in the revised manuscript).

Response to Reviewer's Comments

Response to Reviewer #1's comments

The 3 referees seem to give a similar appreciation of the manuscript : it is well written and gives a high quality ARPES view of the electronic structure of this interesting kagome material, but it fails a bit short of giving a really unexpected and significant progress into the physics of these materials. In their answer, the authors can only repeat why they think kagome physics is interesting in general. I largely agree with them, but I still think that a paper in Nature Communications should go a bit beyond this. I would rather recommend this manuscript for a more specialized journal.

I appreciate the addition of unsymmetrized FS and sublattice decomposition of the band structure, in the supplementary data of the revised manuscript

We thank Reviewer #1 for reviewing our paper again. We also thank the Reviewer for reiterating that our paper is well written and gives a high quality ARPES view of the electronic structure of this interesting kagome material.

As an experimental paper, our paper is significant in a number of aspects: 1. CsTi_3Bi_5 is a new kagome compound recently synthesized which is isostructural to the celebrated AV_3Sb_5 family but with a different effective doping. Therefore, our first ARPES investigation is significant to understand its electronic structure and provide more complete information to understand the physics of Kagome lattice. 2. We have observed coexistence of multi-sets of nontrivial band structures in one single kagome superconductor. In particular, the expected flat band has been clearly observed that lies close to the Fermi level in 135 systems (ATi_3Bi_5 and AV_3Sb_5). The simultaneous existence of such multi-sets of nontrivial band structures not only provides good opportunities to study related physics in the kagome lattice but also new systems to realize novel quantum phenomena. 3. We have demonstrated that the position of the flat band, as well as the van Hove singularities, Dirac cones and the nodal loops, can be tuned over a large energy range by manipulating the chemical composition of the 135 kagome compounds. Our results point to a way to move these characteristic electronic structures to the Fermi level by further adjusting the composition and doping of the 135 systems that will significantly affect superconductivity and produce exotic physical properties.

For materials with a relatively weak electron correlations, the band structure calculation is expected to give a faithful description as demonstrated by many published work on topological materials. Therefore, the agreement between the experiments and band structure calculations should not weaken the significance of our work and affect its publication in high profile journals.

As pointed out by Reviewer #3, our work is clearly interesting to the condensed matter community as evidenced by several manuscripts about ATi_3Bi_5 which recently appeared on arXiv.

Response to Reviewer #3's comments

The changes to the manuscript have made the motivation of the study stronger, but overall there is still no clear sense of why any of the features are important to the material's properties or what is the novelty on a technical or conceptual level. I do not find the other revisions substantial enough to recommend the publication.

In particular, while I agree that the band structure calculation is expected to give a faithful description of a relatively low correlation material, I think the lack of k_z -dependent measurement is a major shortcoming when discussing the discovery of nodal loops in an experimental paper. Any statements in this regard should be softened and should not be presented as a major point in the study, which considerably weakens the overall message.

With that said, I would like to point out that the work is clearly interesting to the condensed matter community as evidenced by several manuscripts which recently appeared on arXiv (while I do understand that the authors had no knowledge of them at the time of original submission, I believe these should also be acknowledged in the revised manuscript).

We thank Reviewer #3 for reviewing our paper again and providing constructive comments and suggestions to further improve our paper. We also thank the Reviewer for pointing out that our work is clearly interesting to the condensed matter community as evidenced by several manuscripts which recently appeared on arXiv.

As an experimental paper, our paper is significant in a number of aspects: 1. CsTi_3Bi_5 is a new kagome compound recently synthesized which is isostructural to the celebrated AV_3Sb_5 family but with a different effective doping. Therefore, our first ARPES investigation is significant to understand its electronic structure and provide more complete information to understand the physics of Kagome lattice. 2. We have observed coexistence of multi-sets of nontrivial band structures in one single kagome superconductor. In particular, the expected flat band has been clearly observed that lies close to the Fermi level in 135 systems (ATi_3Bi_5 and AV_3Sb_5). The simultaneous existence of such multi-sets of nontrivial band structures not only provides good opportunities to study related physics in the kagome lattice but also new systems to realize novel quantum phenomena. 3. We have demonstrated that the position of the flat band, as well as the van Hove singularities, Dirac cones and the nodal loops, can be tuned over a large energy range by manipulating the chemical composition of the 135 kagome compounds. Our results point to a way to move these characteristic electronic structures to the Fermi level by further adjusting the composition and doping of the 135 systems that will significantly affect superconductivity and produce exotic physical properties. We believe that our work

deserves to be published in Nature Communications because of its novelty, high quality data, significance and broad interest.

Following the Reviewer's suggestion, in the revised manuscript, we softened our statements on the Dirac nodal lines. In the abstract, we changed "We also identified the type-II Dirac nodal loops around the Brillouin zone center, the type-III Dirac nodal loops around the zone corners and type-III Dirac nodal lines along the k_z direction." into "We identified the type-II and type-III Dirac nodal lines and their momentum distribution in CsTi_3Bi_5 from the band structure calculations and the measured electronic structures are consistent with the calculated results". In the introduction, we changed "We also identified the Dirac nodal loops and nodal lines in three-dimensional momentum space." into "We also identified the Dirac nodal loops and nodal lines in three-dimensional momentum space from the band structure calculations and the measured electronic structures are consistent with the calculated results".

In addition, following the Reviewer's suggestion, we added more discussion about the Dirac nodal lines in the main text as "We have identified the existence of Dirac nodal lines and their momentum distribution in CsTi_3Bi_5 from our detailed band structure calculations (Fig. 3a-3b and Fig. S5). We used laser-ARPES to measure CsTi_3Bi_5 by taking the advantage of its high instrumental resolution. All the measured electronic structures are in a good agreement with the band structure calculations. Specifically, the measured results of the nodal loops at the k_z from our laser-ARPES measurements (Fig. 3c-3e) are also consistent with the calculated results (Fig. 3f-3h). Our band structure calculations and the laser-ARPES measurements have provided major information on the nodal loops in CsTi_3Bi_5 . To fully characterize the nodal lines in CsTi_3Bi_5 , further ARPES work are needed to probe different k_z s by using different photon energies".

We thank the Reviewer for pointing out that several manuscripts about ATi_3Bi_5 recently appeared on arXiv. Following the Reviewer's suggestion, in the revised manuscript, we added some related references about ARPES study of ATi_3Bi_5 (A=K, Rb and Cs).

Summary of Modifications

1. Following Reviewer #3's suggestion, on page 2, line 33, we changed "We also identified the type-II Dirac nodal loops around the Brillouin zone center, the type-III Dirac nodal loops around the zone corners and type-III Dirac nodal lines along the k_z direction." into "We identified the type-II and type-III Dirac nodal lines and their momentum distribution in CsTi_3Bi_5 from the band structure calculations and the measured electronic structures are consistent with the calculated results".

2. Following Reviewer #3's suggestion, on page 4, line 73, we changed "We also identified the Dirac nodal loops and nodal lines in three-dimensional momentum space." into "We also identified the Dirac nodal loops and nodal lines in three-dimensional momentum space from the band structure calculations and the measured electronic structures are consistent with the calculated results".
3. Following Reviewer #3's suggestion, on page 8, line 227, we added more discussion as "We have identified the existence of Dirac nodal lines and their momentum distribution in CsTi₃Bi₅ from our detailed band structure calculations (Fig. 3a-3b and Fig. S5). We used laser-ARPES to measure CsTi₃Bi₅ by taking the advantage of its high instrumental resolution. All the measured electronic structures are in a good agreement with the band structure calculations. Specifically, the measured results of the nodal loops at the k_z from our laser-ARPES measurements (Fig. 3c-3e) are also consistent with the calculated results (Fig. 3f-3h). Our band structure calculations and the laser-ARPES measurements have provided major information on the nodal loops in CsTi₃Bi₅. To fully characterize the nodal lines in CsTi₃Bi₅, further ARPES work are needed to probe different k_z s by using different photon energies".
4. Following the Reviewer's suggestion, in the revised manuscript, on page 11, line 318, we added "Note added: After the submission of our paper, we became aware of related ARPES work on ATi₃Bi₅ (A=K, Rb and Cs)[70–72]."

REVIEWERS' COMMENTS

Reviewer #3 (Remarks to the Author):

While I appreciate that the authors have softened their statements somewhat, I would argue that this necessary revision weakens the overall conclusion of the manuscript. I agree with Reviewer 1 in that the manuscript should be published in a more specialised journal rather than in Nature Communications.

Response to Reviewer's Comments

Response to Reviewer #3's comments

While I appreciate that the authors have softened their statements somewhat, I would argue that this necessary revision weakens the overall conclusion of the manuscript. I agree with Reviewer 1 in that the manuscript should be published in a more specialised journal rather than in Nature Communications.

We thank the Reviewer for reviewing our paper again. We believe that our work deserves to be published in Nature Communications because of its novelty, high quality data, significance and broad interest.